# Clinical Characteristics and Prognosis of Older Patients with Coronavirus Disease 2019 Requiring Mechanical Ventilation

**DOI:** 10.3390/jpm14060657

**Published:** 2024-06-19

**Authors:** Green Hong, Da Hyun Kang, Sunghoon Park, Su Hwan Lee, Onyu Park, Taehwa Kim, Hye Ju Yeo, Jin Ho Jang, Woo Hyun Cho, Song I Lee

**Affiliations:** 1Division of Allergy, Pulmonary, and Critical Care Medicine, Department of Internal Medicine, Chungnam National University Hospital, Chungnam National University School of Medicine, Daejeon 35015, Republic of Korea; ihongreen@naver.com (G.H.); ibelieveu113@naver.com (D.H.K.); 2Department of Pulmonary, Allergy, and Critical Care Medicine, Hallym University Sacred Heart Hospital, Anyang 14068, Republic of Korea; f2000tj@naver.com; 3Division of Pulmonology, Department of Internal Medicine, Severance Hospital, Yonsei University College of Medicine, Seoul 03722, Republic of Korea; hihogogo@naver.com; 4College of Nursing, Research Institute of Nursing Science, Pusan National University, Yangsan 50612, Republic of Korea; oun0856@naver.com; 5Division of Allergy, Pulmonary, and Critical Care Medicine, Department of Internal Medicine, Pusan National University Yangsan Hospital, Pusan National University School of Medicine, Busan 50612, Republic of Korea; taehwagongju@naver.com (T.K.); dugpwn@naver.com (H.J.Y.); jjhteen1@naver.com (J.H.J.); popeyes0212@hanmail.net (W.H.C.)

**Keywords:** COVID-19, mechanical ventilation, older adults

## Abstract

An older age is associated with severe progression and poor prognosis in coronavirus disease 2019 (COVID-19), and mechanical ventilation is often required. The specific characteristics of older patients undergoing mechanical ventilation and their prognostic factors are largely unknown. We aimed to identify potential prognostic factors in this group to inform treatment decisions. This retrospective cohort study collected data from patients with COVID-19 at 22 medical centers. Univariate and multivariate Cox regression analyses were performed to assess factors that influence mortality. We allocated 434 patients in geriatric (≥80 years) and elderly (65–79 years) groups. The former group scored significantly higher than the elderly group in the clinical frailty scale and sequential organ failure assessment, indicating more severe organ dysfunction. Significantly lower administration rates of tocilizumab and extracorporeal membrane oxygenation and higher intensive care unit (ICU) and in-hospital mortality were noted in the geriatric group. The factors associated with ICU and in-hospital mortality included high creatinine levels, the use of continuous renal replacement therapy, prone positioning, and the administration of life-sustaining treatments. These results highlight significant age-related differences in the management and prognosis of critically ill older patients with COVID-19. Increased mortality rates and organ dysfunction in geriatric patients undergoing mechanical ventilation necessitate age-appropriate treatment strategies to improve their prognoses.

## 1. Introduction

The coronavirus disease 2019 (COVID-19) pandemic has created unprecedented challenges to healthcare systems around the world, with older adults being particularly vulnerable to serious consequences [1]. The prognosis for older patients is often poor due to factors such as diminished immune responses and pre-existing health conditions [2,3]. In addition, the prognosis in this group is influenced by age-related factors such as comorbidities [4], a high degree of frailty [5], and high scores in the sequential organ failure assessment (SOFA) [6]. Understanding the specific clinical outcomes and factors that influence prognosis in older patients is critical for improving treatment strategies and for the optimal allocation of healthcare resources.

Research has consistently shown that an older age is a poor prognostic factor in COVID-19 [7,8], although interindividual variability regarding outcomes is considerable. In particular, geriatric patients—usually defined as individuals 80 years and older—may feature different clinical characteristics and outcomes than relatively elderly patients—usually between 65 and 79 years. For example, geriatric patients often experience more severe respiratory failure [9], a higher incidence of delirium [10], and an increased dependence on interventions such as mechanical ventilation [9]. In addition, the prevalence of pre-existing conditions such as cerebrovascular disease [11] and frailty [12] tends to be higher in this group, which may complicate the management of COVID-19 and negatively affect prognosis.

This study aimed to explore these differences by focusing on geriatric patients with COVID-19 requiring mechanical ventilation compared with elderly, less ill patients. By examining the differences in clinical characteristics and outcomes between these two groups, we aimed to discover important age-related differences that could inform approaches to personalized treatment and management.

## 2. Methods

### 2.1. Study Design

This was a multicenter, retrospective cohort study that reviewed the medical records of adult patients aged 65 years and older who were diagnosed with COVID-19. The evaluation included records from January 2020 to August 2021 at 22 healthcare facilities that treated critically ill patients with COVID-19 (Appendix A). Data from patients who were treated with mechanical ventilation were analyzed (Figure 1).

The principal investigator, Professor Woo Hyun Cho of Pusan National University Yangsan Hospital, oversaw the study, and researcher Onyu Park was responsible for data collection and quality management. Abstractors involved in data collection were thoroughly trained to ensure the consistency and accuracy of data abstraction. Quality control measures were implemented to verify the integrity and reliability of the data collected.

We collected data from the electronic medical records on the enrolled patients’ ages, underlying comorbidities, scores using clinical frailty scales, laboratory findings, therapeutic agents administered, treatments performed in the intensive care unit (ICU) and during continuous renal replacement therapy (CRRT), outcomes, and lengths of ICU and hospital stays. Patient severity was assessed using the SOFA score at the time of initial mechanical ventilation.

COVID-19 was diagnosed as acute SARS-CoV-2 infection via nucleic acid amplification tests or antigen tests on specimens collected from the upper respiratory tract. Critically ill patients with COVID-19 were those who presented with severe symptoms and required intensive medical care [13]. This includes patients with acute respiratory distress syndrome, septic shock, or multiple organ dysfunction who often require ICU admission and mechanical ventilation. Invasive mechanical ventilation was used for patients with severe COVID-19 who were unable to maintain oxygen saturation above 90% using a high-flow nasal cannula device or had unstable vital signs. Restrictions on life-sustaining treatment, such as do-not-resuscitate orders and the completion of forms related to withdrawal or withholding of life-sustaining treatment at any time during hospitalization, were defined as issues of life-sustaining treatment. But specific data on the timing of these decisions to initiate life-sustaining treatment were not collected.

This study was approved by the Institutional Review Board (No. 2021-12-044), and the requirement for informed consent was waived due to the retrospective study design. The collected data were compiled into Excel files and then consolidated at the main coordinating center. The data collection phase was completed in August 2022. Data queries were then addressed, and the final dataset was locked in February 2023. The study then proceeded with the research phase using the curated dataset.

### 2.2. Statistical Analyses

The Statistical Package for the Social Sciences software (SPSS; version 25.0; IBM Corporation, Summers, NY, USA) was used for the statistical analysis. The patients were allocated to geriatric (80 years and older) and elderly (65–79 years) groups. All categorical variables are expressed as percentages, and continuous variables are shown as the median (interquartile range [IQR]: 25–75th percentile) and mean ± standard deviation. The chi-squared test or Fisher’s exact test was used to analyze continuous or categorical data. Cox regression analysis was performed to assess risk factors for ICU and in-hospital mortality. Factors with a *p*-value < 0.1 in the univariate analysis were identified and included in the multivariate analysis. The risk interval for this analysis was defined from the time of ICU admission until either the occurrence of the event (death) or the end of the study period for censored patients. Patients were censored at the time of ICU discharge or at the study period’s end if death did not occur, ensuring the inclusion of those who survived beyond the study or those who were discharged alive. The assumptions of the Cox model were verified, with proportional hazards tested via Schoenfeld residuals showing no significant departures, and linearity of continuous variables assessed using the log hazard function. Non-linear relationships were addressed by categorizing continuous variables or applying appropriate transformations. Factors associated with mortality were expressed as odds ratios (ORs) and 95% confidence intervals (CIs). Statistical significance was defined as *p*-value < 0.05.

## 3. Results

### 3.1. The Characteristics of the Study Patients

A total of 434 older patients with COVID-19 who required mechanical ventilation were enrolled in this study. Of the enrolled patients, 67.7% (*n* = 294) were aged 65–79 years, and 32.3% (*n* = 140) were very old patients aged 80 years or older.

Table 1 shows the baseline characteristics of the enrolled patients. The mean ages were 84.5 ± 3.9 and 71.9 ± 4.3 years in the geriatric and elderly groups, respectively. Male representation was lower in the geriatric group than in the elderly group (50.0% vs. 60.9%, *p* = 0.032). The average body mass index (BMI) was lower in the geriatric patients than in the elderly patients (23.5 ± 4.0 vs. 24.7 ± 3.8, *p* = 0.002). Compared to the elderly group, the geriatric group featured a significantly higher clinical frailty scale (4.4 ± 1.8 vs. 3.0 ± 1.4, *p* < 0.001) and higher SOFA scores (8.3 ± 3.4 vs. 7.5 ± 3.2, *p* = 0.019), indicating more severe organ failure. The prevalence of cardiovascular diseases was significantly higher among the very old patients than in the elderly patients (22.1% vs. 11.6%, *p* = 0.004). Additionally, chronic neurological diseases were notably more common in the geriatric group than in the elderly group (25.7% vs. 12.2%, *p* < 0.001).

Appendix A presents the patients’ initial values of vital signs and laboratory findings. The recorded initial vital signs reveal that the geriatric patients had lower diastolic blood pressure and Glasgow Coma Scale scores, and the laboratory results indicate lower hemoglobin, pH, and higher blood urea nitrogen, creatinine, and lactate levels than the elderly patients.

### 3.2. Treatment Options and Clinical Outcomes

Table 2 shows the treatment regimens and clinical outcomes of the patients. Remdesivir and steroids were commonly administered to both groups without significant differences in the frequency of administration. However, tocilizumab was administered less frequently in the geriatric group (2.9% vs. 7.8%, *p* = 0.045), and fewer geriatric patients received extracorporeal membrane oxygenation (ECMO; 5.7% vs. 17.3%, *p* = 0.001). ICU mortality was significantly higher in the geriatric group (52.1% vs. 38.1%, *p* = 0.006). Significant differences were also noted regarding in-hospital mortality, with a higher rate being observed in geriatric patients (57.1% vs. 40.8%, *p* = 0.001). The duration of mechanical ventilation and length of hospital stay were not significantly different between the groups. Post-discharge clinical frailty was significantly higher in the geriatric group (6.0 ± 1.9 vs. 4.8 ± 1.8, *p* < 0.001), and more patients in this group had the issue of life-sustaining treatments (42.9% vs. 29.9%, *p* = 0.008).

### 3.3. Factors Associated with ICU and In-Hospital Mortality

Table 3 lists the factors associated with ICU mortality. The univariate analysis showed that age, BMI, SOFA score, high creatinine and lactate levels, tocilizumab use, CRRT use, and the issue of life-sustaining treatment were associated with ICU mortality. In the multivariate analysis, high creatinine levels (OR, 1.162; 95% CI, 1.017–1.327; *p* = 0.027), high lactate levels (OR, 1.045; 95% CI, 1.006–1.086, *p* = 0. 025), the use of CRRT (OR, 1.516; 95% CI, 1.037–2.218; *p* = 0.032), and the issue of life-sustaining treatment (OR, 3.580; 95% CI, 2.482–5.164; *p* < 0.001) were associated factors.

Table 4 presents the factors associated with in-hospital mortality. The univariate analysis identified significant associations with age, BMI, SOFA score, high creatinine and lactate levels, the use of CRRT, prone positioning, and an issue of life-sustaining treatment. The multivariate analysis highlighted several significant predictors of in-hospital mortality: high creatinine levels (OR, 1.120; 95% CI, 0.997–1.257; *p* = 0.056), the use of CRRT (OR, 1. 825; 95% CI, 1.266–2.630; *p* = 0.001), prone positioning (OR, 0.592; 95% CI, 0.418–0.840, *p* = 0.003), and the issue of life-sustaining treatment (OR, 5.565; 95% CI, 3.890–7.961; *p* < 0.001).

## 4. Discussion

This study examined in detail the clinical characteristics, treatment strategies, and outcomes in elderly and geriatric patients with COVID-19 who required mechanical ventilation, focusing on the differences in prognosis and health care needs between the two age groups. The analysis showed that geriatric patients—aged 80 years and older—had a more severe clinical profile and higher mortality rates than the elderly patients (65–79 years). These findings highlight the importance of age-appropriate treatment approaches and resource allocation in the management of COVID-19 in older adults, which should reflect the vulnerabilities and specific care needs of geriatric patients.

Our results show that geriatric patients complicate healthcare management and significantly contribute to poorer outcomes. These patients experienced more severe organ dysfunction, as indicated by their higher clinical frailty and SOFA scores, and had a higher prevalence of cardiovascular and neurological diseases. These results are also consistent with the existing literature that shows an advanced age to be associated with severe respiratory infections, including COVID-19 [14,15]. An advanced age is associated with a very high prevalence of multimorbidity, with approximately 81% of the patients in this age group having at least one comorbidity, such as hypertension, diabetes, or cardiovascular disease [16]. Likewise, other studies have reported that older patients generally have a greater burden of comorbidities than younger populations [17,18]. These conditions emphasize the challenges patients face in managing COVID-19, highlighting the complexity of their healthcare demands and the need for an integrated care approach that addresses exacerbating underlying conditions.

There are challenges in managing geriatric patients with reduced physiological capacity to tolerate aggressive therapy. Disparities in the administration of therapies, such as tocilizumab and ECMO, reflect these challenges. While the restricted use of these aggressive therapies in geriatric patients reflects the careful clinical judgment of the potential risks and benefits, specific studies linking age to limited COVID-19 therapies are scarce [19,20]. Previous studies have shown that an advanced age is often associated with decisions to limit life-sustaining treatment [21,22], suggesting that similar considerations may influence less aggressive treatment approaches for geriatric patients with COVID-19. These associations underscore the need for nuanced clinical decision making that takes into account the unique complexities of treating critical illness in the older population.

The significantly higher ICU and in-hospital mortality rates observed in the geriatric population indicate that specialized care strategies tailored to the needs of these patients should be provided. Extensive research supports these observations. For example, the results of a systematic review and meta-analysis published in *BMJ Global Health* [23] highlighted a clear association between an older age and severe COVID-19 outcomes, including increased hospitalization rates, ICU admission, and mortality. Other studies have also shown that an older age is not only associated with a more severe initial clinical presentation but also with a higher likelihood of progression to severe disease [24] and increased mortality [7,8]. These patterns underscore the importance of strategic care planning and resource allocation to effectively manage the care trajectory of geriatric patients after hospitalization. Given the similar resource-intensive needs of the elderly cohort, and despite the lack of significant differences regarding the duration of mechanical ventilation and length of stay, proactive and predictive care strategies are paramount to optimizing the outcomes in this vulnerable population.

Our multivariate analysis identified high creatinine and lactate levels, the use of CRRT, and the issue of life-sustaining treatment as significant predictors of mortality. These factors highlight the important role of renal dysfunction and metabolic acidosis in the prognosis of patients with severe COVID-19. Our findings are consistent with those in previous studies that indicated that acute kidney injury increases the need for mechanical ventilation and COVID-19 severity [25,26]. In addition, our data corroborate observations of shock responses and multiorgan failure in patients receiving CRRT [27,28], highlighting the influence of these adverse effects on COVID-19 prognosis. The strong correlation between the issue of life-sustaining treatment and mortality underscores the importance of initiating early and clear discussions about treatment goals, particularly in older populations. In support of this approach, previous research has indicated that considerations regarding life-sustaining treatment significantly influence prognosis [20]. In addition, although not evaluated in this study, an older age is considered a significant risk factor for severe outcomes from COVID-19 [29,30], with chronic disease [4,31], a higher degree of frailty [12], a higher SOFA score [6,32], and elevated levels of inflammatory markers [33,34] generally being associated with worse clinical outcomes. These complexities call for a comprehensive management strategy that addresses both the direct effects of COVID-19 and the broader underlying health issues that affect older patients, especially those requiring intensive interventions such as CRRT.

This study had several limitations. First, the retrospective design may have introduced bias related to data collection and patient selection. To overcome this limitation, the data were collected by experienced ICU physicians or research nurses. Second, because this study was conducted at multiple centers, differences in treatment protocols and philosophies may have influenced the results. These disparities are unlikely to have affected the study findings because all centers implemented the same protocol for ventilator use and prophylaxis bundle and applied the same treatment approach for COVID-19. Third, data on ventilator settings were lacking. We attempted to collect data on the patients’ initial positive end-expiratory pressure, tidal volume, and other parameters, but the information was inconsistent, and many values were not available; therefore, these variables could not be used in the analysis. In future studies, we believe that the inclusion of ventilator settings in the analysis will help guide treatment decisions.

## 5. Conclusions

Our study highlights the complex interplay among age, comorbidities, treatment decisions, and clinical outcomes in older patients with COVID-19 who required mechanical ventilation. The findings presented here underscore the need for age-specific treatment protocols that consider the unique clinical profiles and prognoses of elderly and geriatric patients to optimize outcomes and the efficient use of healthcare resources.

## Figures and Tables

**Figure 1 jpm-14-00657-f001:**
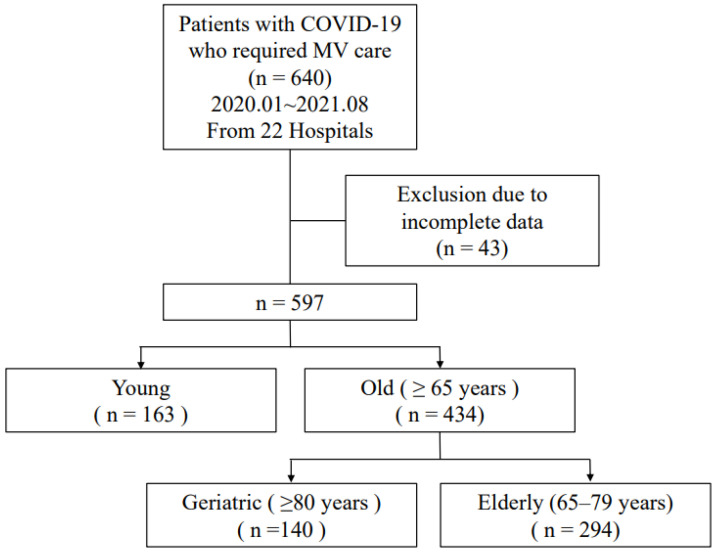
Study flow diagram of patients with COVID-19 undergoing mechanical ventilation (MV).

**Table 1 jpm-14-00657-t001:** Baseline characteristics of enrolled patients.

Variables	Total Patients (*n* = 434)	Elderly (*n* = 294)	Geriatric (*n* = 140)	*p*-Value
Age	76.0 ± 7.2	71.9 ± 4.3	84.5 ± 3.9	<0.001
Male (%)	249 (57.4)	179 (60.9)	70 (50.0)	0.032
Body mass index	24.3 ± 3.9	24.7 ± 3.8	23.5 ± 4.0	0.002
Clinical frailty scale	3.5 ± 1.7	3.0 ± 1.4	4.4 ± 1.8	<0.001
SOFA score	7.8 ± 3.3	7.5 ± 3.2	8.3 ± 3.4	0.019
Vaccination history	19 (4.4)	16 (5.4)	3 (2.1)	0.116
Comorbidity (%)				
Hypertension	289 (66.6)	195 (66.3)	94 (67.1)	0.866
Diabetes	171 (39.4)	113 (38.4)	58 (41.4)	0.551
Cardiovascular disease	65 (15.0)	34 (11.6)	31 (22.1)	0.004
Chronic lung disease	46 (10.6)	27 (9.2)	19 (13.6)	0.165
Chronic neurological disease	72 (16.6)	36 (12.2)	36 (25.7)	<0.001
Chronic kidney disease	40 (9.2)	24 (8.2)	16 (11.4)	0.272
Chronic liver disease	9 (2.1)	5 (1.7)	4 (2.9)	0.429
Hematologic malignancy	6 (1.4)	5 (1.7)	1 (0.7)	0.411
Solid organ tumor	32 (7.4)	23 (7.8)	9 (6.4)	0.603

Data are presented as mean ± standard deviation or number (%), unless otherwise indicated. SOFA: Sequential Organ Failure Assessment.

**Table 2 jpm-14-00657-t002:** Treatment and clinical outcomes.

Variables	Total Patients (*n* = 434)	Elderly (*n* = 294)	Geriatric (*n* = 140)	*p*-Value
Treatment (%)				
Remdesivir	291 (67.1)	203 (69.0)	88 (62.9)	0.200
Steroid	417 (96.1)	283 (96.3)	134 (95.7)	0.785
Tocilizumab	27 (6.2)	23 (7.8)	4 (2.9)	0.045
Convalescent plasma	19 (4.4)	12 (4.1)	7 (5.0)	0.662
Intervention in the ICU				
CRRT	93 (21.4)	56 (19.0)	37 (26.4)	0.080
Prone positioning	134 (30.9)	98 (33.3)	36 (25.7)	0.108
ECMO	59 (13.6)	51 (17.3)	8 (5.7)	0.001
Tracheostomy	152 (35.0)	103 (35.0)	49 (35.0)	0.994
Outcomes				
Duration of mechanical ventilation	14.0 (8.0–33.5)	15.0 (8.0–33.0)	14.0 (7.0–36.0)	0.533
ICU mortality	185 (42.6)	112 (38.1)	73 (52.1)	0.006
ICU LOS	23.0 (14.0–42.0)	25.0 (15.0–42.3)	20.0 (11.0–40.0)	0.920
In-hospital mortality	200 (46.1)	120 (40.8)	80 (57.1)	0.001
Hospital LOS	31.0 (19.5–57.0)	33.0 (21.0–56.3)	26.0 (16.0–59.0)	0.640
Cause of death				
Respiratory failure	119 (27.4)	75 (25.5)	44 (31.4)	0.196
Septic shock c MOF	61 (14.1)	36 (12.2)	25 (17.9)	0.116
Cardiac death	6 (1.4)	2 (0.7)	4 (2.9)	0.069
Neurologic death	1 (0.2)	1 (0.3)	0 (0)	0.490
Others ^†^	13 (3.0)	6 (2.0)	7 (5.0)	0.091
Post-discharge clinical frailty scale (*n* = 234)	5.1 ± 1.9	4.8 ± 1.8	6.0 ± 1.9	<0.001
Issue of life-sustaining treatment	148 (34.1)	88 (29.9)	60 (42.9)	0.008

Data are presented as mean ± standard deviation or number (%), unless otherwise indicated. ^†^ Death of undetermined etiology. ICU: intensive care unit, CRRT: continuous renal replacement therapy, ECMO: extracorporeal membrane oxygenation, LOS: length of stay, MOF: multi-organ failure.

**Table 3 jpm-14-00657-t003:** Univariate and multivariate risk factors associated with ICU mortality. (Cox regression analysis).

	Univariate Analysis			Multivariate Analysis		
	OR	95% CI	*p*-Value	OR	95% CI	*p*-Value
Age	1.029	1.009–1.048	0.004	1.009	0.986–1.032	0.462
Male	1.091	0.811–1.467	0.565			
Body mass index	1.040	1.003–1.079	0.034	1.016	0.971–1.064	0.485
Clinical frailty scale	1.030	0.943–1.124	0.516			
SOFA score	1.041	0.996–1.088	0.073	0.994	0.940–1.050	0.817
Comorbidity						
Cardiovascular disease	1.027	0.682–1.546	0.899			
Chronic neurological disease	1.021	0.703–1.484	0.912			
Hematologic malignancy	0.680	0.217–2.134	0.509			
Solid organ tumor	1.258	0.753–2.104	0.381			
Initial vital sign						
Diastolic BP, mmHg	0.993	0.983–1.003	0.164			
GCS	1.006	0.967–1.046	0.783			
Laboratory findings						
White blood cell, 10^3^/µL	1.001	0.989–1.014	0.839			
Albumin, g/dL	0.917	0.693–1.213	0.545			
Creatinine, mg/dL	1.172	1.071–1.282	0.001	1.162	1.017–1.327	0.027
C-reactive protein, mg/dL	1.000	0.999–1.002	0.407			
P/F ratio, mmHg	0.999	0.998–1.001	0.328			
Lactate, mmol/L	1.072	1.035–1.111	<0.001	1.045	1.006–1.086	0.025
Treatment						
Tocilizumab	1.631	0.961–2.769	0.070	1.029	0.367–2.890	0.956
CRRT	2.174	1.617–2.922	<0.001	1.516	1.037–2.218	0.032
Prone positioning	0.801	0.586–1.093	0.162			
Issue of life-sustaining treatment	3.924	2.882–5.342	<0.001	3.580	2.482–5.164	<0.001

OR: Odds Ratio, CI: Confidence Interval, SOFA: Sequential Organ Failure Assessment, GCS: Glasgow Coma Scale, CRRT: Continuous Renal Replacement Therapy.

**Table 4 jpm-14-00657-t004:** Univariate and multivariate risk factors associated with in-hospital mortality. (Cox regression analysis).

	Univariate Analysis			Multivariate Analysis		
	OR	95% CI	*p*-Value	OR	95% CI	*p*-Value
Age	1.039	1.020–1.058	<0.001	1.016	0.995–1.038	0.140
Male	1.054	0.794–1.400	0.716			
Body mass index	1.039	1.004–1.075	0.030	1.024	0.981–1.069	0.281
Clinical frailty scale	1.019	0.937–1.109	0.656			
SOFA score	1.045	1.001–1.091	0.045	1.004	0.952–1.059	0.871
Comorbidity						
Cardiovascular disease	0.949	0.646–1.393	0.789			
Chronic neurological disease	1.102	0.773–1.570	0.592			
Hematologic malignancy	0.689	0.220–2.160	0.523			
Solid organ tumor	1.320	0.830–2.100	0.240			
Initial vital sign						
Diastolic BP, mmHg	0.995	0.985–1.004	0.286			
GCS	1.005	0.967–1.045	0.801			
Laboratory findings						
White blood cell, 10^3^/µL	1.005	0.993–1.016	0.443			
Albumin, g/dL	0.883	0.682–1.144	0.346			
Creatinine, mg/dL	1.172	1.079–1.273	<0.001	1.120	0.997–1.257	0.056
C-reactive protein, mg/dL	1.001	0.999–1.002	0.294			
P/F ratio, mmHg	0.999	0.998–1.001	0.445			
Lactate, mmol/L	1.034	1.003–1.066	0.034	0.998	0.963–1.033	0.902
Treatment						
Tocilizumab	1.471	0.869–2.491	0.151			
CRRT	2.461	1.845–3.282	<0.001	1.825	1.266–2.630	0.001
Prone positioning	0.7586	0.559–1.021	0.068	0.592	0.418–0.840	0.003
Issue of life sustaining treatment	5.394	3.989–7.295	<0.001	5.565	3.890–7.961	<0.001

OR: Odds Ratio, CI: Confidence Interval, SOFA: Sequential Organ Failure Assessment, GCS: Glasgow Coma Scale, CRRT: Continuous Renal Replacement Therapy.

## Data Availability

All data generated or analyzed during this study are included in this published article and its Appendix A.

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
