# Peer review of "Clinical Characteristics and Prognosis of Older Patients with Coronavirus Disease 2019 Requiring Mechanical Ventilation"

_jpm, 2024, doi:10.3390/jpm14060657_

Round 1

Reviewer 1 Report

Comments and Suggestions for Authors

Thank you for the opportunity to review this manuscript. I'd like to commend the authors for conducting this study. Some comments are provided below to strengthen the validity and robustness of this study.

1. Since it is a retrospective study, it is better to explain the data collection process (data abstraction, abstractor, ICD codes use, etc.)

2. The authors performed the Cox regression analysis, so please describe in detail about the risk interval, censoring, assumptions of Cox PH regression.

3. It would be more informative to illustrate a study flow figure.

4. [Line 82-86] Please consider adding or editing this variable since the time of life-sustaining treatments play a vital role in mortality and disease progression.

Comments on the Quality of English Language

Moderate editing of English language required.

Reviewer 2 Report

Comments and Suggestions for Authors

Major

This is a retrospective cohort study from 22 medical centers in Korea aiming at identifying prognostic factors that influence mortality. Patients were divided in two groups i.e. very old (80 years and older) and younger old (65-79 years). Data analysis showed statistically significant difference in factors influencing mortality among the two groups.

 Minor

We suggest renaming the two groups as geriatric (80 years and older) and elderly (65-79 years) patients throughout the Manuscript. A comment on the specific needs of geriatric population in the era of Personalized Medicine at the Discussion section would be welcome.  

Comments on the Quality of English Language

Recommendation: More compact paragraphs with main finding as first sentence. 

Reviewer 3 Report

Comments and Suggestions for Authors

First of all, thank you to the authors for this comprehensive study. In the study, clinical characteristics of COVID-19 patients were examined retrospectively. However, the results analyzed and presented have been shown and known in many previous studies. Therefore, the study lacks innovation and its limited contribution to the literature.

Round 2

Reviewer 3 Report

Comments and Suggestions for Authors

Good job.